# Dietary Supplementation with *Saccharomyces cerevisiae*, *Clostridium butyricum* and Their Combination Ameliorate Rumen Fermentation and Growth Performance of Heat-Stressed Goats

**DOI:** 10.3390/ani11072116

**Published:** 2021-07-16

**Authors:** Liyuan Cai, Jiangkun Yu, Rudy Hartanto, Desheng Qi

**Affiliations:** 1Department of Animal Nutrition and Feed Science, College of Animal Science and Technology, Huazhong Agricultural University, Wuhan 430070, China; doriacai@mail.hzau.edu.cn (L.C.); yjk555@webmail.hzau.edu.cn (J.Y.); rudyhartanto@lecturer.undip.ac.id (R.H.); 2Department of Animal Science, Faculty of Animal and Agricultural Sciences, Diponegoro University, Semarang 50275, Indonesia

**Keywords:** goats, heat stress, *Saccharomyces cerevisiae*, *Clostridium butyricum*, rumen fermentation, growth performance

## Abstract

**Simple Summary:**

Heat stress commonly happens to goats raised in the Jianghuai region of China during the summer and inevitably contributes to a loss of animal production. Dietary supplementation with probiotics is one of the practical approaches to improve animal production. By dietary supplementation with *Saccharomyces cerevisiae*, *Clostridium butyricum*, and their combination to the heat-stressed goats, the present study showed these probiotics effectively alleviate heat stress by improving the rumen fermentation function and growth performance. Therefore, this study provides a reference for the application of these two probiotics in ruminant production during heat stress.

**Abstract:**

This study aimed to evaluate the effects of *Saccharomyces cerevisiae*, and their combination on rumen fermentation and growth performance of heat-stressed goats. Twelve heat-stressed goats (20.21 ± 2.30 kg) were divided equally into four groups: control group (CG, fed the basal diet, *Saccharomyces cerevisiae* supplemented group (SC, 0.60% *Saccharomyces cerevisiae* added to the basal diet), *Clostridium butyricum* supplemented group (CB, 0.05% *Clostridium butyricum* added to the basal diet), and their combination supplemented group (COM 0.60% *Saccharomyces cerevisiae* and 0.05% *Clostridium butyricum* added to the basal diet) and were assigned to a 4 × 3 incomplete Latin square design. The rumen fluid and feces were collected for fermentation parameters and feed digestibility analysis, and animal growth performance was also assessed during all the experiment periods. The results showed that rumen pH, rumen cellulolytic enzymes (avicelase, CMCaes, cellobiase, and xylanase) activities, and the concentrations of rumen total volatile fatty acid (TVFA), acetic acid, and propionic acid were significantly increased with *Saccharomyces cerevisiae*, *Clostridium butyricum*, and their combination supplementation (*p* < 0.05). Besides, the dry matter intake (DMI), average daily gain (ADG), and the digestibility of dry matter (DM), neutral detergent fiber (NDF), and acidic detergent fiber (ADF) were significantly increased (*p* < 0.05) with supplemented these probiotics. However, the ammonia nitrogen (NH_3_-N) concentration only significantly increased in CB and A/P ratio (acetic acid to propionic acid ratio) only significantly increased in SC and CB. These results indicated that the supplementation with these probiotics could ameliorate rumen fermentation and growth performance of heat-stressed goats.

## 1. Introduction

Ruminants exhibit low tolerance to heat because rumen fermentation produces large amounts of heat [1]. Heat stress brings about various adverse effects to rumen functions, include decreasing ruminal pH, affecting the microbial composition, lowering the production of rumen TVFA, decreasing the digestibility of nutrients, and causing oxidative stress [1,2,3,4,5]. Thus, it decreases the production performance of ruminants and brings economic loss to the goat breeding industry [6]. 

Probiotics have been widely used in ruminants and non-ruminants to improve feed digestion, performance, and health status [7]. Yeast is one of the probiotics commonly applied in ruminant nutrition research and production. The ability of yeast to consume oxygen contributes to the maintenance of an oxygen-free environment in the rumen, thus facilitating the growth and reproduction of anaerobic rumen microbes, especially the majority of cellulolytic bacteria [8,9]. Supplementation with yeast culture was also reported to improve the concentrations of NH_3_-N and TVFA, and the digestibilities of DM, NDF, and ADF of dairy cows [10]. So far, few studies were focused on the application of yeast on heat-stressed ruminants, and usually inconsistent results were obtained. Dai et al. (2009) found active dry yeast reduced the rectal temperature of the heat-stressed cows and prolonged their peak milk production [11]. However, no effect was observed on the yield of milk and DMI by feeding an *Saccharomyces cerevisiae* culture to mid-lactation dairy cows during the summer, although the feed efficiency was improved [12]. *Clostridium butyricum* is a strictly anaerobic endospore-forming Gram-positive butyric acid-producing bacterium and is a promising probiotic candidate [13]. Most previous studies of *Clostridium butyricum* were conducted on monogastric animals and poultry, few studies on ruminants were reported. Previous studies showed that *Clostridium butyricum* could improve the production performance of weaned piglets and chickens [14,15]. Additionally, supplementation with *Clostridium butyricum* of 2.5 × 10^5^ CFU/kg to the weaned piglets, the ADG and feed conversion ratio were significantly increased [16]. In poultry production, *Clostridium butyricum* not only improved chicken production performance but also improved the fertilization rate of eggs [13]. Besides, diet supplemented with 0.2% *Clostridium butyricum* increased the egg-laying rate and fertilization rate by 28.58% and 4.10%, respectively [14,17]. For ruminants, *Clostridium butyricum* has the potential to improve rumen fermentation and degradability, possibly through their metabolites, which could increase the number of rumen bacteria as other probiotics [18]. Nevertheless, the effectiveness of *Clostridium butyricum* on ruminants to alleviate heat stress is still inconclusive.

Therefore, the objectives of this study were to evaluate the effects of *Saccharomyces cerevisiae*, *Clostridium butyricum* and their combination on rumen fermentation and growth performance of Chinese crossbred goats under heat stress conditions. This study will provide a scientific reference for alleviating the adverse effects of heat stress on ruminants in the practical production.

## 2. Materials and Methods

### 2.1. Animals, Diet, and Management

Twelve female Macheng black × Boer crossbred goats were obtained from Boda Animal Husbandry Science and Technology Development Co. Ltd. (Hefei City, Anhui Province, China). These goats were aged 6.0 ± 1.0 months with a body weight of 20.21 ± 2.30 kg were kept in natural ventilation house with individual pens. These goats were fed twice a day at 8:00 a.m. and 17:00 p.m. with free access to water. The ingredients and nutritional composition of the diet are given in Table 1. Vaccination and other prophylactic measures were implemented with procedures described by Vatta et al. [19]. This study was conducted from June to September and was approved by the Animal Care and Use Committee of Huazhong Agricultural University (Approval code HZAUGO-2015-007).

### 2.2. Probiotics Feeding Experimental Design 

The modeling processes of heat-stressed goats were described by Cai et al. [5]. In brief, an air conditioner and an air heater were used to control room temperature of goat house at 33.2 ± 2.7 °C. Water was sprinkled on the ground as needed to ensure the relative humidity was 74.4 ± 2.3%. The temperature–humidity index (THI) was used as an indicator of heat stress in goats and was calculated as
THI = db°F − ((0.55 − 0.55 RH) (db°F − 58)),
where db °F is the dry bulb temperature (°F) and RH is the relative humidity (%). In this thermal environment, the THI was 87.0 and the goats suffered from heat stress. The goats were kept in this thermal environment for two weeks. The heat stressed goat modeling was established. Heat-stressed goats were then kept in this thermal environment for the present study. Twelve heat-stressed goats were divided equally into four groups and assigned to a 4 × 3 incomplete Latin square design. The groups were as follows: control group (CG, fed the basal diet, *Saccharomyces cerevisiae* supplemented group (SC, 0.60% *Saccharomyces cerevisiae* added to the basal diet), *Clostridium butyricum* supplemented group (CB, 0.05% *Clostridium butyricum* added to the basal diet), and their combination supplemented group (COM, 0.60% *Saccharomyces cerevisiae* and 0.05% *Clostridium butyricum* added to the basal diet) *Saccharomyces cerevisiae* was obtained from Angel Yeast Co., Ltd. (Yichang, China) and had a content of 2.0 × 10^10^ CFU/g. *Clostridium butyricum* live cell product was obtained from Huijia Biotechnology Co., Ltd. (Huzhou, China) at 1.0 × 10^8^ CFU/g. Three experimental cycles were included in this study. The design of groups in each cycle was shown in Table 2. Each experimental cycle lasted for 20 days.

In each experimental cycle, 5 g Chromium oxide (Cr_2_O_3_) was taken as external digestibility marker added to the diet of days 17 to 19 to determine of digestibility of nutrients. Between experimental cycles, all of the goats were fed a basal diet for 15 days to eliminate the influence of previous probiotic treatment and prepare for the next experimental cycle. Rumen fluid was collected on the last day of each experimental cycle and the methods of rumen fluid collection and pretreatment were described by Cai et al. [5]. In brief, rumen fluid was collected on the last day of each experimental cycle by a flexible stomach tube with a vacuum pump (Jin Teng GM-0.33A, Tianjin, China) 4 h after morning feed. Next, the rumen fluid was strained through four layers of gauze to remove big feed particles and then transferred the filtrate to CO_2_-containing bottles to maintain anaerobic conditions. The filtrate was immediately stored at −20 °C for further analysis. Fecal samples were collected from the rectum of each goat before the morning and afternoon feedings at the last three days of each experimental cycle. Additionally, fecal samples from the same treatment were pooled and stored at −20 °C for further analysis. 

### 2.3. Measurements

The pH and oxidation-reduction potential (ORP) values were measured immediately after the rumen fluid was collected. Rumen fluid was centrifuged at 12,000× *g* for 15 min at 4 °C, and the supernatants were used for ammonia nitrogen (NH_3_-N) and volatile fatty acids (VFA) analysis. NH_3_-N concentration was determined using spectrophotometry as described by Maitisaiyidi et al. [20]. The VFA concentration was determined by the gas chromatography described by Yang et al. [21]. In brief, 0.20 mL supernatant was added to 1.00 mL 25% (*w/v*) metaphosphoric acid and centrifuged at 10,000 r/min for 10 min. Then, the supernatant was injected into Chrompack CP-Wax 52 fused silica column (30 m × 0.53 mm × 1.00 μm) of gas chromatography equipped with flame ionization detector (Model 2010, Shimazu, Japan). The activities of avicelase, hydrolytic enzyme (CMCase), cellobiase, and xylanase in rumen fluid were determined as described by Wang and Wang [22]. The DMI and body weight of heat-stressed goats were measured within each experimental cycle. DMI was calculated by subtracting the weight of food refused from the weight of that offered on the previous day. The body weight of goats was measured by an electronic weighing balance (PS-2000 Platform Scale, Salter Brecknell, Fairmont, MN, USA) in the morning before offering feed and water. The body weights were recorded at the start and end of each experimental cycle for ADG calculations. Feed and fecal samples were analyzed as described by the AOAC [23] official methods for DM (method #930 15). NDF and ADF were determined as described by Zhang et al. [24]. All analyses were carried out in triplicate to ensure the accuracy of the test results.

### 2.4. Statistical Analysis

Microsoft Excel 2016 (Microsoft, Redmond, WA, USA) was used for data collection and to calculate average values and standard errors. All the data, including rumen fermentation, activities of enzymes, and animal growth performance data were analyzed in R packages (v4.0.5, GitHub Inc., San Francisco, CA, USA) with two-way analysis of variance (ANOVA) tests followed by post hoc Dunn test for each significant factor or interaction. *p* values of less than 0.05 were considered statistically significant.

## 3. Results

### 3.1. Rumen Fermentation Parameters of Heat-Stressed Goats with Probiotic Supplements 

The ruminal pH was significantly increased, while the rumen ORP was significantly decreased in SC, CB, and COM compared with that of the CG (*p* < 0.05). Meanwhile, the concentration of NH_3_-N was significantly increased (*p* < 0.05) in SC and CG compared with that of CG and COM. Additionally, the concentrations of TVFA, acetic acid, propionic acid, and A/P ratio were noticeably increased (*p* < 0.05) in SC and CB compared with that of CG and COM. The activities of rumen avicelase, CMCase, cellobiase, and xylanase in SC, CB, and COM were significantly increased (*p* < 0.05) compared with that in rumen of CG. Besides, the activities of rumen CMCase and xylanase in CB was significantly higher than that of SC and COM (*p* < 0.05). Ruminal fermentation parameters and the of ruminal cellulolytic enzyme activities of heat-stressed goats with probiotics supplement are shown in Table 3.

### 3.2. Growth Performance of Heat-Stressed Goats with Probiotic Supplements 

The DMI, ADG, and digestibilities of DM, NDF, and ADF were significantly increased (*p* < 0.05) in SC, CB, and COM compared with that of CG. Additionally, CB exhibited a greater effect in enhas ncing the digestibility of DMI and DM owing to the higher digestibility in CB compared with SC and COM (*p* < 0.05). The growth performance parameters of heat-stressed goats with probiotics supplement are shown in Table 4.

## 4. Discussion

Supplementation with probiotics was able to alleviate the adverse effects of heat stress in livestock production. A previous study reported that dietary supplementation with yeast resulted in a higher ruminal pH in cows [25,26]. In the present study, the ruminal pH was increased with *Saccharomyces cerevisiae, Clostridium butyricum,* and their combination supplementation, which is similar to the results of previous studies on cows with live yeast supplementation during the hot season [27]. These results suggested that *Saccharomyces cerevisiae* could effectively alleviate the pH shift caused by heat stress [5]. This result should be ascribed to yeast is able to produce some metabolites, such as vitamin and organic acids, which can promote the growth and reproduction of lactic acid utilizing bacteria [9]. Moreover, yeast could enhance other microbes to compete with lactic acid producing bacteria for soluble sugars, and then reduce lactic acid production [26]. Therefore, yeast has the function for stabling rumen pH. In contrast, several studies reported that supplementation with *Saccharomyces cerevisiae* could decrease ruminal pH [28] or have no effect on the pH [29,30]. The discrepancies could attribute to the different sources or strains of this probiotic applied in different studies. Few studies have investigated the effects of *Clostridium butyricum* on rumen fermentation. It is reported that calves were adapted to a 50% high-concentrate diet for 1 week, and then *Clostridium butyricum* was given to the calves once daily for five days at 1.5 or 3.0 g/100 kg body weight. As a result, both doses of it improved the reduction in the 24 h mean ruminal pH in the calves [31]. In this study, heat-stressed goats feeding with *Clostridium butyricum* alone or combining with *Saccharomyces cerevisiae* increased the ruminal pH. The stabilizing/increasing effect of probiotics on ruminal pH might be attributed to the activity enhancement of some predominant rumen bacteria to consume lactate [31]. Moreover, probiotics can promote the abundance of rumen protozoa, which could also lower the ruminal lactic acid concentration [32]. The rumen ORP reflects the activity of the microbiota and the amounts of reducing substances, and provides another perspective for fermentation processes in the rumen. It has been reported that supplementation with *Saccharomyces cerevisiae* at a concentration of 1.3 mg·mL^−1^ increased the oxygen disappearance rate by 46–89% and decreased the rumen ORP of sheeps [33]. Live yeast was also discovered to be a balancer of rumen fluid ORP and effective in reducing rumen ORP [34]. In the current study, probiotics supplementation could significantly decreased the rumen ORP in the rumen, which is in accordance with previous studies. The ability of *Saccharomyces cerevisiae* and *Clostridium butyricum* to consume oxygen in the rumen and on the surface of feedstuffs could primarily contributed to the decrease of ruminal ORP. Previous studies found that yeast supplementation led to a significant decrease in the concentration of NH_3_-N in the rumen [9,28,35], similar result was obtained in this study. The increase in the NH_3_-N concentration could ascribe to the rumen microbiota promoted by probiotics supplementation to degrade and hydrolyze protein [35]. However, a previous study also showed that yeast had no effect on the NH_3_-N concentration in the rumen of cows [26]. The inconsistent results perhaps result from variations in the feeding system, animal species, age and physiological state of the ruminants, frequency of feeding, doses of yeast, and composition of the diets, and environmental conditions in different studies. In the present study, supplementation of *Clostridium butyricum* or the combination significantly raise the concentration of NH_3_-N in the rumen of heat-stressed goats. Further confirmation is needed since few studies have been conducted on *Clostridium butyricum*. The results of previous studies on the effects of *Saccharomyces cerevisiae* on rumen TVFA were inconsistent. Studies with active dry yeast or *Saccharomyces cerevisiae* supplemented to sheep and cattle reported significant increase in the concentration of TVFA in the rumen of sheep [7,12,34,36,37,38]. However, a study has shown that supplementing of *Saccharomyces cerevisiae* did not alter the TVFA concentration in the rumen [39]. Diets supplemented with *Saccharomyces cerevisiae* were shown to increase propionic acid production and A/P ratio in the rumen [12,37]. Few studies investigated the effects of *Clostridium butyricum* on VFA production in the rumen. It was reported that calves fed with *Clostridium butyricum* at 1.5 or 3.0 g/100 kg body weight in the diet did not affect the ruminal VFA concentrations [31]. In the present study, *Saccharomyces cerevisiae*, *Clostridium butyricum*, and their combinations supplementation improve the concentration of TVFA, acetic acid, and propionic acid concentrations, and the A/P ratio. The increase in the TVFA concentration could be attributed to the ability of yeast to stimulate the activities of rumen microbes, especially fibrolytic bacteria [34,40,41,42]. The mechanisms underlying the enhancement of the TVFA concentration by *Clostridium butyricum* are likely to be similar to those of *Saccharomyces cerevisiae*. In addition, in this study, the A/P ratio following probiotic supplementation was increased significantly relative to the control ratio, indicating that the rumen fermentation mode may be altered by these probiotic supplementations. The increase of the A/P ratio usually means a decrease of fermentation efficiency, and the increase of this ratio of in this study is due to the increase extent of acetic acid more than that of propionic acid. These results indicated that these probiotics are beneficial to rumen fermentation of heat-stressed goats. Some researchers have considered that the change in VFA concentration caused by supplementation with probiotics is not worthy of attention because it will disappear once probiotic supplementation is terminated. This issue should be taken into account in production practices.

A previous study found that the digestibility of DM in sheep was improved by supplementation with *Saccharomyces cerevisiae* [30]. Lila et al. [39] reported that NDF digestibility was increased by 10.5% with 5 g/day of yeast supplemented to goat kids. Similarly, in this study, *Saccharomyces cerevisiae* effectively increased the digestibilities of DM, NDF, and ADF. Previous studies scarcely evaluated the effects of *Clostridium butyricum* on rumen digestibilities of DM, NDF and ADF. As a good probiotic resource [42], it is important to evaluate the role of *Clostridium butyricum* in rumen digestibilities of DM, NDF and ADF. In the current study, it is found that live *Clostridium butyricum* improved the digestibilities of rumen DM, NDF and ADF of heat-stressed goats. In addition, *Saccharomyces cerevisiae* and *Clostridium butyricum* are good sources of vitamins and minerals [43,44] which also ameliorate the growth and reproduction of rumen cellulolytic bacteria and fungi, may therefore improve fiber digestion. In this study, supplementation with *Saccharomyces cerevisiae*, *Clostridium butyricum* or their combination significantly increased the DMI of heat-stressed goats. This result is similar to that of a previous study showed *Saccharomyces cerevisiae* increased feed intake in the early lactation stage of lactating dairy goats with 0.2 g/day of yeast supplementation [45]. However, another study showed that supplemented with *Saccharomyces cerevisiae* had no effect on the DMI of cows under summer heat stress conditions [25]. There are few studies performed on the effect of *Clostridium butyricum* on DMI of ruminants. Besides, our study showed that *Saccharomyces cerevisiae*, *Clostridium butyricum**,* or their combination improved the ADG of goats, which is similar to the results of studies on cows and goats with *Saccharomyces cerevisiae* supplementation [46,47,48]. However, other studies that supplemented with *Saccharomyces cerevisiae* alone or a combination of it and *L. sporogenes* did not affect ADG of animals. In addition, it has been reported that the use of *Clostridium butyricum* in the diet (2.5 × 10^8^ cfu/kg feed) of weaning piglets and chickens can improve weight gain and feed efficiency [16]. Similarly, *Clostridium butyricum* was found to have positive effects on the growth performance of broiler chickens [42]. Although *Saccharomyces cerevisiae* and *Clostridium butyricum* can have beneficial effects on growth performance, their effects may vary from different studies. This variation can be attributed to factors such as variation in basal diets (hay, straw, and forage), the number of live cells of probiotics, dosage, and feeding strategies [49]. The activities of cellulolytic enzymes (avicelase, CMCase, cellobiase and xylanase) were improved in the present study, which could be the main reason for the increased digestibilities of DM, NDF, and ADF. However, the effects on the activities of these enzymes varied among supplementation levels, which might result from the different effects of these probiotics on bacteria that produce the various cellulolytic enzymes. The results of this study suggest that supplementation with *Saccharomyces cerevisiae* and *Clostridium butyricum* alleviated the adverse effects of heat stress on the activities of cellulolytic enzymes.

## 5. Conclusions

The *Saccharomyces cerevisiae*, *Clostridium butyricum*, and their combination supplemented to the diet ameliorate rumen conditions by increasing pH and decreasing ORP, and enhance the rumen fermentation functions by increasing digestibility of nutrients and improve the VFA production, and thereafter improve the growth production of heat-stressed goats. Therefore, supplementation with these probiotics can be an effective measure to alleviated adverse effects of heat stress on goats.

## Figures and Tables

**Table 1 animals-11-02116-t001:** Ingredients and nutrition (g/kg) of the basic diet fed to the goats.

Ingredient	Content
Alfalfa	562
Ground corn	264
Soybean meal	84
Wheat barn	73
Ca_2_HPO_4_	7
Premix *	10
Nutrition Level	
Dry matter	951
Organic matter	854
Crude protein	173
Neutral detergent fibre	434
Acid detergent fibre	257
Ca	5.9
P	3.2

* Premix contained per kg: 20.70 g Mg, 0.50 g Fe, 1 g Mn, 2 g Zn, 43 mg Se, 47 mg I, 54 mg, Co, 90,000 IU vitamin A, 17,000 IU vitamin D, 1750 IU vitamin E.

**Table 2 animals-11-02116-t002:** 4 × 3 incomplete Latin-Square design of the experiment.

Groups	P1	P2	P3
T0	basal diet	basal diet + SC	basal diet + CB
T1	basal diet + SC	basal diet	basal diet + combination
T2	basal diet + CB	basal diet + combination	basal diet + SC
T3	basal diet + combination	basal diet + CB	basal diet

P1–P3 represent the experimental cycle; SC, CB, and combination represent *Saccharomyces cerevisiae, Clostridium butyricum*, and their combination, respectively.

**Table 3 animals-11-02116-t003:** Ruminal fermentation parameters and the of ruminal cellulolytic enzyme activities of heat-stressed goats with probiotics supplementation.

Parameters	Treatment				SEM	*p* Value		
	CG	SC	CB	COM		SC	CB	SC × CB
pH	6.58 ^a^	6.72 ^b^	6.70 ^b^	6.73 ^b^	0.04	<0.001	<0.001	<0.001
ORP (mV)	−161.3 ^a^	−171.0 ^b^	−183.4 ^b^	−177.1 ^b^	7.13	0.045	<0.001	0.01
NH_3_-N (mg 100 mL^−1^)	9.20 ^a^	10.87 ^ab^	12.12 ^b^	9.81 ^a^	0.57	0.25	0.01	0.04
Acetic acid (mmol L^−1^)	19.38 ^a^	28.12 ^b^	30.77 ^b^	21.59 ^a^	2.78	0.007	<0.001	<0.001
Propionic acid (mmol L^−1^)	14.08 ^a^	18.2 ^b^	20.27 ^b^	13.72 ^a^	1.64	0.002	0.02	<0.001
Butyric acid (mmol L^−1^)	12.38	12.80	14.67	11.69	1.66	0.061	0.400	0.056
A/P ratio	1.38 ^a^	2.13 ^b^	1.57 ^b^	1.52 ^a^	0.81	0.008	0.05	0.223
Avicelase (IU mL^−1^)	1.31 ^a^	1.55 ^b^	1.82 ^b^	1.61 ^b^	0.02	0.050	<0.001	<0.001
CMCaes (IU mL^−1^)	1.36 ^a^	2.58 ^b^	3.11 ^c^	2.57 ^b^	0.01	<0.001	<0.001	<0.001
Cellobiase (IU mL^−1^)	2.44 ^a^	4.46 ^b^	4.71 ^b^	4.53 ^b^	0.05	<0.001	<0.001	<0.001
Xylanase (IU mL^−1^)	4.54 ^a^	6.40 ^b^	7.31 ^c^	5.62 ^b^	0.10	<0.001	<0.021	0.043

^a–c^ Means within a row with different superscripts letters differ significantly (*p* < 0.05); with no and the samesuperscripts letters indicate no significant difference (*p* > 0.05) in same row.

**Table 4 animals-11-02116-t004:** The growth performance parameters of heat-stressed goats with probiotics supplementation.

Parameters	Treatment				SEM	*p* Value		
	CG	SC	CB	COM		SC	CB	SC × CB
DMI (kg)	0.79 ^a^	0.84 ^b^	0.87 ^c^	0.84 ^b^	0.04	0.005	< 0.001	<0.001
ADG (kg)	0.08 ^a^	0.19 ^b^	0.12 ^b^	0.12 ^b^	0.01	0.040	< 0.001	0.004
Digestibilities of								
DM (%)	50.58 ^a^	60.84 ^b^	66.46 ^c^	65.44 ^b^	3.63	0.001	<0.001	<0.001
NDF (%)	38.32 ^a^	51.04 ^b^	54.13 ^b^	52.20 ^b^	3.59	<0.001	<0.001	<0.01
ADF (%)	37.82 ^a^	50.03 ^b^	50.06 ^b^	49.29 ^b^	3.00	<0.001	<0.001	<0.01

^a–c^ Means within a row with different superscripts letters differ significantly (*p* < 0.05); with no and the same superscripts letters indicate no significant difference (*p* > 0.05) in same row.

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
