# Peer review of "Dietary Supplementation with Saccharomyces cerevisiae, Clostridium butyricum and Their Combination Ameliorate Rumen Fermentation and Growth Performance of Heat-Stressed Goats"

_animals, 2021, doi:10.3390/ani11072116_

Round 1
Reviewer 1 Report
Comments for animals 1269465:
The authors of the presented manuscript present a study in which growing goats are supplemented with probiotics in the context of heat stress. The probiotics display a beneficial effect on rumen fermentation and animal performance parameters.
Major remarks:
1) Revision of language by a native speaker is strongly recommended. Only a few examples from the first lines are listed as I want to focus on the content of the manuscript and language editing is not my expertise.
- The reviewing process already starts badly. The title of the article does not seem to be correct: The Saccharomyces cerevisiae and Clostridium butyricum are benefitial for rumen fermentation and growth performance of heat-stressed goats
- The simple summary contains language flaws too:
- There are many ways to alleviate heat stress, and feeding probiotics is one of the most effective ways.
- Here, we found that the rumen fermentation and the growth performance of goats were improved by Saccharomyces cerevisiae, Clostridium butyricum, and their combination.
- …
- Such examples occur in all parts of the manuscript.
2) The usage of different abbreviations for the same feeding group (COM/HG) is unfortunate and should be unified.
3)The “washout” period of 5 days seems really short when it comes to rumen microbiome adaptations.
Minor remarks:
l.56 CB is a member of the microbial taxon Bacillus. Change to bacterium
Thoroughly check for typos (e.g. l.193 eumen -> rumen, l.2ßß ruemn->rumen)
l.213 “and so on”àthis is unfortunate wording
Critical questions:
- Why was the control group not cycled in the incomplete latin square while 5-day periods were applied to remove probiotic effects of previous cycles anyway? (l.98-l.106) If 5 days were enough to wash out effects of prior cycles, control group could be cycled (see major remark (3)).
- Why was the interval for daily weight gain chosen in a manner that the 5-day preparation periods were included? (l. 127-l129)
- Why did the authors choose t-tests? Does data fulfill requirements (e.g. normality)? Why was ANOVA with posthoc tests not used?
Author Response
Response to Reviewer 1 Comments
Point 1: Revision of language by a native speaker is strongly recommended. Only a few examples from the first lines are listed as I want to focus on the content of the manuscript and language editing is not my expertise.
The reviewing process already starts badly. The title of the article does not seem to be correct: The Saccharomyces cerevisiae and Clostridium butyricum are beneficial for rumen fermentation and growth performance of heat-stressed goats
The simple summary contains language flaws too:
There are many ways to alleviate heat stress, and feeding probiotics is one of the most effective ways.
Here, we found that the rumen fermentation and the growth performance of goats were improved by Saccharomyces cerevisiae, Clostridium butyricum, and their combination.
…
Such examples occur in all parts of the manuscript.
Response 1: At your suggestion, the manuscript was revised by a native speaker. The title of this manuscript was changed to “Dietary supplementation with Saccharomyces cerevisiae, Clostridium butyricum and their combination ameliorate rumen fermentation and growth performance of heat-stressed goats” (at line 2-4).
Point 2: 2) The usage of different abbreviations for the same feeding group (COM/HG) is unfortunate and should be unified.
Response 2: “HG” was replaced by is “ COM” throughout this manuscript.
Point 3: 3)The “washout” period of 5 days seems really short when it comes to rumen microbiome adaptations.
Response 3: We are sorry to write “15” as “5” after double checked our original experiment record. Now “5” was replaced 15 (at line 116).
Minor remarks:
Point 4: l.56 CB is a member of the microbial taxon Bacillus. Change to bacterium
Response 4: “Bacillus” was change to “bacterium” (at line 64).
Point 5: Thoroughly check for typos (e.g. l.193 eumen -> rumen, l.2ßß ruemn->rumen)
Response 5: the spelling mistakes were revised thoroughly.
Point 6: l.213“and so on”àthis is unfortunate wording
Response 6: “and so on”was deleted and the sentence was rewritten. (at line 224-227).
Point 7: Critical questions: Why was the control group not cycled in the incomplete latin square while 5-day periods were applied to remove probiotic effects of previous cycles anyway? (l.98-l.106) If 5 days were enough to wash out effects of prior cycles, control group could be cycled (see major remark (3)). Why was the interval for daily weight gain chosen in a manner that the 5-day preparation periods were included? (l. 127-l129)
Response 7: We changed our description in the Latin square design especially the group setting for each cycle after discussed with the other authors, I am sorry to have made wrong described on this due to my own incorrect understanding. The 4 × 3 incomplete Latin-Square design was revised in Table 2. We are sorry to write “15” as “5”. Now “5” was replaced with 15 (at line 116). The DMI and ADG were only observed within each experimental cycle days and excluded the 15 days interval. There is no difference between the 5-day preparation period and the 15 day formal feeding period. Therefore, “Each experimental cycle was divided into a 5 days preparation period and a 15 days formal experiment period.” was changed to “Each experimental cycle was last for 20 days.” (at line 110). And the 5-day preparation periods were included. We have also made targeted modifications according to your questions in the manuscript (line 140-141).
Point 8: Why did the authors choose t-tests? Does data fulfill requirements (e.g. normality)? Why was ANOVA with posthoc tests not used?
Response 8: There were some mistakes in the statistical method. And the correct statistical methods were used to process these data statistics again. Two-way ANOVA with posthoc tests were used for Statistical analysis. (at line 151-155).
Reviewer 2 Report
Overall:
The manuscript “The Saccharomyces cerevisiae and Clostridium butyricum are benefit for rumen fermentation and growth performance of heat-stressed goats” reports the use of fungi as probiotics in improving ruminal fermentation and growth performance of goats during heat stress.
The title should be adjusted; remove “The” and replace “benefit” with “beneficial”.
The affiliation number 2 is not associated with any of the authors. The English language should be revised throughout the entire manuscript.
Abstract:
Line 24: no probiotics – control group?
Line 25: it is the first time the abbreviations are being introduced and therefore the full meaning should be given.
Line 26: assigned instead of designed…
Line 29: again, first time abbreviations should have the full meaning…
Re-write this section. Best to state a brief description of M&M and sampling analysis
Introduction:
Please revise language throughout this section. Do not focus so much in studies in monogastrics (line 58-63).
Line 40: Reference not linked (1)
Line 43: provide reference
Line 47: which activities? Elaborate…
Line 49: NDF instead of ND,F
Line 50: provide references
Line 56: how is it a good probiotic? What do you mean here?
Line 66: elaborate on the results of this study and specific mechanisms
Materials and Methods:
Line 85: Which vaccines/prophylactic measures were those?
Line 88: Show data vertically and separate well diet ingredients and nutrient composition
Line 92: Twelve… How did you induce heat stress?
Line 93: How did you choose these concentrations?
Line 98: three
Line 102: why did you give the group T0 always the same diet? Why didn´t you alternate this one too?
Line 105: 5 days preparation period consisted of what?
Line 106: subscripts in the chemical formulas…
Line 108: Was there any carryover effect? How can you be sure that 5 days were enough?
Line 114: How did you sample with a O2-free environment?
Line 119: Define ORP… it´s the first time it shows up
Line 121: give some info, briefly describe method, same for line 131
Line 135: average values of the triplicates?
Line 138: which variables did you test?
Line 139: data normally distributed and of equal variance? how did you account for animal variability and run effect?
Results:
Line 143: All vs all or treatments vs CON? statistical significance defined at which level?
Line 153: Give overall P-value and not only between means comparisons
Line 154: ORP, unit?
Line 155-156: these sentences makes no sense. Please re-write
Line 159: Give overall P-value
Discussion:
Line 169: you haven´t tested any other…
Line 174: why is this beneficial?
Line 175: You did not test this, as all animals were under heat-stress conditions
Line 178: Give overall p-value and move this table to before the discussion.
Line 189: which predominant rumen bacteria consume lactate?
Line 193: “rumen” instead of “eumen”
Line 200: rumen…
Line 204: all animals were under heat stress, which adverse effects do you mean?
Line 210: just one study is cited…
Line 220: what does your study show?
Line 226: the proportion…?
Line 233: you didn´t analyse microbial composition
Line 236: again, you don´t know this…
Line 239: future research
Line 262: italicize bacteria/fungi species names
Conclusion:
Elaborate…
Author Response
Response to Reviewer 2 Comments
Point 1: The title should be adjusted; remove “The” and replace “benefit” with “beneficial”.
The affiliation number 2 is not associated with any of the authors. The English language should be revised throughout the entire manuscript.
Response 1: The title of this manuscript was changed to “Dietary supplementation with Saccharomyces cerevisiae, Clostridium butyricum and their combination ameliorate rumen fermentation and growth performance of heat-stressed goats” (at line 1-4). The affiliation number 2 should be associated with Rudy Hartanto, we are sorry to miss the mark to the author. (at line 5)
Point 2: Line 24: no probiotics – control group?
Response 2: We change no probiotics to the control group (CG) fed with the basal diet to avoid any misunderstanding. (at line 22).
Point 3: Line 25: it is the first time the abbreviations are being introduced and therefore the full meaning should be given.
Response 3: According to your suggestion, we have revised it at line 30, 32-35.
Point 4: Line 26: assigned instead of designed…
Response 4: “designed” was replaced by “assigned”. (at line 25)
Point 5: Line 29: again, first time abbreviations should have the full meaning…
Response 5: We double-checked this point and made corresponding revisions. (at line 30, 32-35).
Point 6: Re-write this section. Best to state a brief description of M&M and sampling analysis
Response 6: According to your suggestion, we have rewritten the “Abstract”. (at 19-36)
Point 7: Please revise language throughout this section. Do not focus so much in studies in monogastrics (line 58-63).
Response 7: According to your suggestion, we asked a language editor to revise our manuscript. Most previous studies on Clostridium butyricum were focused on its functions in monogastric animals and poultry, few studies on ruminants were accessible, we have no choice and will make some comprises.
Point 8: Line 40: Reference not linked (1)
Response 8: We are sorry to have marked this citation in the wrong place. Now it is adjusted to the correct position.(at line 45).
Point 9: Line 43: provide reference
Response 9: a reference ([6]) is now provided at line 49.
Point 10: Line 47: which activities? Elaborate…
Response 10: Well, we rewrote this part and made some changes, now this should not be a problem. (at line 52-55)
Point 11: Line 49: NDF instead of ND,F
Response 11: Spelling mistakes including “ND,F” in this manuscript were carefully corrected ( at line 55 and throughout this manuscript)
Point 12: Line 50: provide references
Response 12: a references was provided at line 57.
Point 13: Line 56: how is it a good probiotic? What do you mean here?
Response 13: We have rewritten this sentence now, new description is a “promising probiotics candidate”(at line 63-64)
Point 14: Line 66: elaborate on the results of this study and specific mechanisms
Response 14: In the cited article, the author only reported that feeding Clostridium butyricum could increased the fertilization rate and egg production rate of, but did not elucidate the mechanism. This is also the deficiency of many studies on the effect of Clostridium butyricum, further study is required to elucidate the possible mechanisms. (at line 71-73)
Point 15: Line 85: Which vaccines/prophylactic measures were those?
Response 15: Vaccination and other prophylactic measures were decribed by Vatta et al. (2007), we added the reference [19] in the section of our manuscript. (at line 91)
Point 16: Line 88: Show data vertically and separate well diet ingredients and nutrient composition
Response 16: The table 1 was adjusted according to your suggestion at line 95.
Point 17: Line 92: Twelve… How did you induce heat stress?
Response 17: The modeling processes of heat-stressed goats were added at line 99. And “12” was replaced by “Twelve” at line 85 and 100.
Point 18: Line 93: How did you choose these concentrations?
Response 18: According to our previous experiments using multiple supplement levels of Saccharomyces cerevisiae, Clostridium butyricum, and their combinations (including in vitro rumen fermentation experiment and feeding experiment, we finally determined the dosage of probiotics in this experiment. As part of the data will be used in another study, we will not mention too much here. Thanks a lot for your understanding.
Point 19: Line 98: three
Response 19: “3” was replaced by “Three” at line 109.
Point 20: Line 102: why did you give the group T0 always the same diet? Why didn´t you alternate this one too?
Response 20: We changed our description in the Latin square design especially the group setting for each cycle after discussed with the other authors, I am sorry to have made wrong described on this due to my own incorrect understanding. The 4 × 3 incomplete Latin-Square design was revised in Table 2. (at line 111)
Point 21: Line 105: 5 days preparation period consisted of what?
Response 21: There is no difference between the 5-day preparation period and the 15 day formal feeding period. Therefore, “Each experimental cycle was divided into a 5 days preparation period and a 15 days formal experiment period.” was changed to “Each experimental cycle was last for 20 days.” (at line 110).
Point 22: Line 106: subscripts in the chemical formulas…
Response 22: “Cr2O3” was replaced by “Cr2O3” at line 114.
Point 23: Line 108: Was there any carryover effect? How can you be sure that 5 days were enough?
Response 23: I'm sorry for writing 15 days as 5 days by mistake, and “5” was replaced by “15” at line 116. To our knowledge, feeding basal diet for 15 days that enough to eliminate the influence of previous treatment.
Point 24: Line 114: How did you sample with a O2-free environment?
Response 24: This inaccurate description has been corrected at line 122-124. (The rumen fluid was strained through four layers of gauze to remove big feed particles, and then transferred the filtrate to CO2-containing bottles to maintain anaerobic conditions.)
Point 25: Line 119: Define ORP… it´s the first time it shows up
Response 25: The full name of ORP was given at line 129.
Point 26: Line 121: give some info, briefly describe method, same for line 131
Response 26: The method was briefly described at line 134-138.
Point 27: Line 135: average values of the triplicates?
Response 27: “All analyses were carried out in triplicate to ensure the accuracy of the test results.” means technical repetition for parameters or data of each sample (at line 148-149).
Point 28: Line 138: which variables did you test?
Response 28: The variables refer to the parameters of rumen fermentation (pH, ORP, NH3-N, and VFA) and growth performance (DMI, ADG, the digestibilities of DM, NDF, and ADF).
Point 29: Line 139: data normally distributed and of equal variance? how did you account for animal variability and run effect?
Response 29: There were some mistakes in the statistical method. And the correct statistical methods were used to process these data statistics again. ( at line 152-155)
Point 30: Line 143: All vs all or treatments vs CON? statistical significance defined at which level?
Response 30: This is a comparison of pH and ORP between the control group and each probiotics supplemented group, statistical significance was defined as the P values less than 0.05. (at 158-159)
Point 31: Line 153: Give overall P-value and not only between means comparisons.
Response 31: We made changes in our results according to your requirement. (in Table 3 and 4; at line 169 and 179)
Point 32: Line 154: ORP, unit?
Response 32: The unit for ORP is “mV”, and the unit was added in Table 3.
Point 33: Line 155-156: these sentences makes no sense. Please re-write
Response 33: The sentences are now rewritten
Point 34: Line 159: Give overall P-value
Response 34: the overall P-value was added in Table 3 and 4. (at line 169 and 179)
Point 35: Line 169: you haven´t tested any other…
Response 35: Based on the previous studies, we described probiotics as one of the effective ways to alleviate heat stress in livestock production.
Point 36: Line 174: why is this beneficial?
Response 36: Because yeast is able to produce some metabolites, such as vitamin and organic acids, which can promote the growth and reproduction of lactic acid utilizing bacteria. And, yeast could enhance other microbes to compete with lactic acid producing bacteria for soluble sugars, and then reduce lactic acid production. Therefore, yeast has the function for stabling rumen pH. (at line191-196 )
Point 37: Line 175: You did not test this, as all animals were under heat-stress conditions
Response 37: This test was carried out in the process of heat-stressed goat modeling, please refer to our previous research. (Cai, L.Y.; Yu, J.K.; Hartanto, R.; Zhang, J. C.; Yang, A.; Qi, D.S. Effects of heat challenge on growth performance, ruminal, blood and physiological parameters of Chinese crossbred goats. Small Rumin Res. 2019, 174, 125-130.). And this reference is also added to line188 as [5].
Point 38: Line 178: Give overall p-value and move this table to before the discussion.
Response 38: The overall P-value was added in table 4 and this table was moved before “discussion section”.(at line 179).
Point 39: Line 189: which predominant rumen bacteria consume lactate?
Response 39: This study did not study the effect of probiotics on the composition and function of rumen microbiota. For the concept of lactic acid utilizing bacteria, it was cited from a previous study. Therefore, we only speculated here, and did not specify the predominant mirobiota. Previous study reported by Ogunade et al. (2019)
that dietary yeast supplementation increased the relative abundance of carbohydrate-fermenting bacteria (such as Ruminococcus albus, R. champanellensis, R. bromii, and R. obeum) and lactate-utilizing bacteria (such as Megasphaera elsdenii, Desulfovibrio desulfuricans, and D. vulgaris).
(Ogunade, I.M.; Lay, J.; Andries, K,; Christina, J.; McManus, Frederick, B.B.Effects of live yeast on differential genetic and functional attributes of rumen
microbiota in beef cattle. J. Anim. Sci. Biotechno. 2019, 10, 68.)
Point 40: Line 193: “rumen” instead of “eumen”
Response 40: Sorry for the spelling mistakes, we have double checked and corrected them now at line 209.
Point 41: Line 200: rumen…
Response 41: “ruemn” was replaced by “rumen” at line 220.
Point 42: Line 204: all animals were under heat stress, which adverse effects do you mean?
Response 42: I mean that the adverse effects of heat stress on rumen fermentation and growth of heat-stressed. This is based on our previous research.
Point 43: Line 210: just one study is cited…
Response 43: More references are added now. (at line 224).
Point 44: Line 220: what does your study show?
Response 44: Our study showed that the TVFA, acetic acid, and propionic acid concentrations, and the A/P ratio were significantly increased with Saccharomyces cerevisiae, Clostridium butyricum, and their combination supplementation compared with control group.
Point 45: Line 226: the proportion…?
Response 45: This is a wrong word and has been deleted.
Point 46: Line 233: you didn´t analyse microbial composition
Response 46: This study did not involve in the effects of probiotics on the composition and function of rumen microbiota, and the speculation of rumen microbial function was based on previous studies.
Point 47: Line 236: again, you don´t know this…
Response 47: This is just a speculation of the related function of rumen microbiota based on previous studies.
Point 48: Line 239: future research
Response 48: Probiotics, as feed additives, have an effect when they are added, but the effect disappears when they are not used based on previous studies. Further studies are needed to elucidate specific mechanisms on this point.
Point 49:Line 262: italicize bacteria/fungi species names
Response 49: “L. sporogenes” was replaced by “L. sporogenes” at line 279.
Point 50: Conclusion: Elaborate…
Response 50: The Conclusion section was rewrote at line 295-300.

Round 2
Reviewer 2 Report
Thanks a lot for your modifications, it has substantially improved the manuscript. You have also answered the majority of my questions.
However, please keep in mind that I am meant to review the current manuscript as it is and not judge on previous publications derived from this same experiment. Therefore, if you want to bring anything from another publication linked to this experiment, please refer to it as your companion paper so that the reviewers and readers are aware of where to look for more information. Additionally, the readers must understand the experiment solely by reading the current manuscript and not by making a collection of all the publications associated with this trial. Having said that, you have to provide more details about methodologies and materials whenever deemed important for the full comprehension of this manuscript.
Some minor points:
Lines 22-24: start by adding the full name of the treatment groups first and only then the acronym. Same goes for the rest of the manuscript where this happens.
Line 89-90: vaccination and prophylactic measures have to be described here… you are citing a manual, could be any measure specified there. What I am interested in knowing is if there were antibiotics administered to the animals or other types of drugs that could have impacted the microbiome and therefore affected your study.
Line 97: Even if this is a companion paper, you have to at least briefly describe the heat-stress induction method.
Lines 148-153: Provide details about method used and how you dealt with run and animal variability. Was the data normally distributed? How was this tested?
Tables 3 and 4. Where do these P-values come from? What does SC*CB mean? Why do you show individual P-values instead of the P-value of the treatment? The question here will be if there was a treatment effect; the differences between treatments vs CG are then given in the superscripts!
Line 182: I have to insist in this point, this sentence needs to be re-written to attest the veracity of your findings e.g., “Supplementation with probiotics was able to alleviate the adverse effects of heat stress in livestock production”. You cannot say “it was one of”, as you have not tested any alternative to probiotics supplementation.
Line 254: Remove And, that is not a correct way to start a sentence
Line 297: an effective measure to
Author Response
Response to Reviewer
Point 1: However, please keep in mind that I am meant to review the current manuscript as it is and not judge on previous publications derived from this same experiment. Therefore, if you want to bring anything from another publication linked to this experiment, please refer to it as your companion paper so that the reviewers and readers are aware of where to look for more information. Additionally, the readers must understand the experiment solely by reading the current manuscript and not by making a collection of all the publications associated with this trial. Having said that, you have to provide more details about methodologies and materials whenever deemed important for the full comprehension of this manuscript.
Response 1: Thank you very much for your comments and suggestion. We are so sorry for our negligence. In the revised manuscript, the process of heat-stressed goat modeling was briefly described.(at line100-109). Through your suggestion, we are also deeply realized that we should pay more attention to the details in our manuscript and strive to add some details to make it easier for readers to understand.
Point 2: Lines 22-24: start by adding the full name of the treatment groups first and only then the acronym. Same goes for the rest of the manuscript where this happens.
Response 2: According to your suggestion, the name of treatment groups start by the full name first and then the acronym throughout this manuscript. (at line 22-25, 111-115)
Point 3: Line 89-90: vaccination and prophylactic measures have to be described here… you are citing a manual, could be any measure specified there. What I am interested in knowing is if there were antibiotics administered to the animals or other types of drugs that could have impacted the microbiome and therefore affected your study.
Response 3: According to the guidance of Vatta et al. (2007) and our goat feeding practice, vaccination and prophylactic measures were applied in this study. Since the crossbreed goats are relatively not susceptible to diseases and have not been ill during our experiment process, we didn’t use other medicine or antibiotics except for vaccine and medicine bath. Quad-components were used for the prevention of Braxy, Struck, Lamb Dysentery, and Enterotoxaemia. Monovalent vaccine was applied for the prevention of Goatpox, Colibacillosis in lambs, Orf, infectious Pleuropneumonia, and Streptococcosis. In this study, a traditional Chinese medicine prescription was used to give medicine bath to goats to drive out ectoparasites. The formula and production process are as follows: Firstly, the Camphor leaves, tobacco, Artemisia argyi, Salix, Urtica cannabina, Akebia, ragwort, Meliae Cortex, Motherwort, mother chrysanthemum, Honeysuckle, Garlic, Zeolite, Sulfur, Alum were prepared according to the weight ratio of 25: 25: 20: 20: 18: 18: 16: 15: 13: 12: 12: 10: 8: 8: 6: 5: 5: 5. Next, water was added to the first 13 herbs for dilution (w/v=1:12, g/ml) before the herb-water mixture was cooked for 40 minutes. Then, Zeolite powder, sulfur and alum were mixed and grinded into fine powder, and garlic was made into mashed garlic. Finally, all the components were mixed together and filtered to obtain the filtrate for bath.
Point 4: Line 97: Even if this is a companion paper, you have to at least briefly describe the heat-stress induction method.
Response 4: The heat-stress induction method was briefly described as follow: An air conditioner and an air heater were used to control room temperature of goat house at 33.2 ± 2.7°C. Water was sprinkled on the ground as needed to ensure the relative humidity was 74.4 ± 2.3%. The temperature-humidity index (THI) was used as an indicator of heat stress in goats and was calculated as THI=db°F-{(0.55-0.55RH) (db°F -58)}, where db °F is the dry bulb temperature (°F) and RH is the relative humidity (%). In this thermal environment the THI was 87.0, the goats were suffered from heat stress. Keep the twelve goats in this thermal environment for two weeks. The heat stressed goat modeling was established. Heat-stressed goats were then kept in this thermal environment for the present study. (at line100-109)
Point 5: Lines 148-153: Provide details about method used and how you dealt with run and animal variability. Was the data normally distributed? How was this tested?
Response 5: As we know, the Latin-Square design is usually applied to increase the sample size (number of experiment repetitions) of each group, thus reduce variations brought about by the animal individuals. Yes, data with normal distribution are required by ANOVA methods regardless of numbers of factors. However, referred to some statisticians online, a big sample size is needed to check the distribution of a dataset, that is the reason that most of studies didn’t test the distribution while using ANOVA for statistical analysis except for some special parameters (e.g. alpha diversity indices of microbial communities). The parameters in the current study are commonly examined by ANOVA referred to the related papers. Our dataset is comprised of two factors (Saccharomyces cerevisiae and Clostridium butyricum) with both two levels (SC: 0, 0.6% DM; CB: 0, 0.05% DM), therefore, two-way ANOVA method is applied to check the significance difference of main effect of two factors and interaction.
Point 6: Tables 3 and 4. Where do these P-values come from? What does SC*CB mean? Why do you show individual P-values instead of the P-value of the treatment? The question here will be if there was a treatment effect; the differences between treatments vs CG are then given in the superscripts!
Response 6: The P-values comes from the two-way ANOVA tests (like Goopy et al., 2014). two-way ANOVA test comes out two main effects and an interaction effect, the SC*CB means interaction effect between the SC and CB factors, unlike one-way ANOVA which only gives one P-value of one factor (the P-value treatment you mentioned). Yes, in two-way ANOVA, we didn’t give a sole treatment effect, although we did the post-hoc test after checking the main effect and interaction effect by treated all the groups as one treatment after two-way ANOVA test to compare difference between groups. And the superscripts come from the post-hoc test. (Goopy, P. J.; Donaldson, A.; Hegarty, R.; Vercoe, E.P.;mHaynes, F. Low-methane yield sheep have smaller rumens and shorter rumen retention time. Brit. J. Nutr. 2004, 111, 578-585.)
Point 7: Line 182: I have to insist in this point, this sentence needs to be re-written to attest the veracity of your findings e.g., “Supplementation with probiotics was able to alleviate the adverse effects of heat stress in livestock production”. You cannot say “it was one of”, as you have not tested any alternative to probiotics supplementation.
Response 7: According your suggestion “Supplementation with probiotics was one of the effective ways to alleviate the adverse effects of heat stress in livestock production.” was replaced by “Supplementation with probiotics was able to alleviate the adverse effects of heat stress in livestock production.”(at line 194-195). After careful consideration, we also think that the proposed expression is more appropriate.
Point 8: Line 254: Remove And, that is not a correct way to start a sentence
Response 8: “And” was removed at line 267.
Point 9: Line 297: an effective measure to
Response 9: “effective measures to” was replaced to “an effective measure to” (at line 310)